# Online Ad Hoc Teamwork under Partial Observability

**Pengjie Gu**[1]**, Mengchen Zhao** [2,*]**, Jianye Hao**[3,2]**, Bo An**[1]
School of Computer Science and Engineering, Nanyang Technological University, Singapore[1]
Noah's Ark Lab, Huawei[2]
College of Intelligence and Computing, Tianjin University[3]
`{pengjie.gu, boan}@ntu.edu.sg, {zhaomengchen, haojianye}@huawei.com`

## Abstract

Autonomous agents often need to work together as a team to accomplish complex cooperative tasks. Due to privacy and other realistic constraints, agents might need to collaborate with previously unknown teammates on the fly. This problem is known as *ad hoc teamwork*, which remains a core research challenge. Prior works usually rely heavily on strong assumptions like full observability, fixed and predefined teammates' types. This paper relaxes these assumptions with a novel reinforcement learning framework called ODITS, which allows the autonomous agent to adapt to arbitrary teammates in an online fashion. Instead of limiting teammates into a finite set of predefined types, ODITS automatically learns latent variables of teammates' behaviors to infer how to cooperate with new teammates effectively. To overcome partial observability, we introduce an information-based regularizer to derive *proxy* representations of the learned variables from local observations. Extensive experimental results show that ODITS significantly outperforms various baselines in widely used ad hoc teamwork tasks.

## 1 Introduction

Recently, autonomous agents including robotics and software agents are being widely deployed in different environments. In many tasks, they are progressively required to cooperate with other unknown teammates on the fly. For example, in case of search and rescue tasks in a disaster, due to privacy or lack of time, deployed robots need to interact with robots from other companies or laboratories, whose coordination protocols might not be explicitly provided in advance (Barrett and Stone, 2015). Besides, in the domain of game AI (Yannakakis, 2012), virtual agents are required to assist different agents controlled by human players. To effectively complete these tasks, autonomous agents must show high adaptation ability to collaborate with intrinsically diverse and unknown teammates. This problem is known in the literature as *ad hoc teamwork* (Stone et al., 2010).

Existing approaches on ad hoc teamwork usually assume that all teammates' behaviors are categorized into several *predefined* and *fixed* types, which corresponds to different coordination strategies (Barrett and Stone, 2015; Durugkar et al., 2020; Mirsky et al., 2020). Then, by reasoning over the type of interacting teammates, the agent switches its behavior to the corresponding policy. If the types are correctly recognized and the strategies are effective, the agent would accomplish the given cooperation task well. However, defining sufficiently descriptive types of teammates requires prior domain knowledge, especially in uncertain and complex environments. For example, in human-AI collaboration in Hanabi (Bard et al., 2020), there are often a wide variety of cooperative behaviors showed by human players. It is challenging for predefined types to cover all possible human players' behaviors. Further, teammates' strategies might be rapidly evolving throughout the entire teamwork. If the agent assumes that teammates' behavioral types are static and cannot adapt to current teammates' behaviors in an *online* fashion, teamwork would suffer from serious miscoordination (Ravula et al., 2019; Chen et al., 2020). Rescue and search tasks are an essential class of such examples (Ravula et al., 2019). On the other hand, existing techniques (Barrett and Stone, 2015; Albrecht and Stone, 2017; Chen et al., 2020; Ravula et al., 2019) try to utilize Bayesian posteriors over teammate types to obtain optimal responses. To effectively compute posteriors, they usually assume that the agent could always know other teammates' observations and actions. However, this

---

*Corresponding author

assumption is unrealistic in partial observable environments, in which each agent is not aware of other agents' observations.

To address the issues mentioned above, this paper introduces an adaptive reinforcement learning framework called **O**nline a**D**aptation via **I**nferred **T**eamwork **S**ituations (ODITS). Our key insight is that teamwork performance is jointly affected by the autonomous agent and other teammates' behaviors. So, the agent's optimal behavior depends on the current *teamwork situation*, which indicates the influence on the environmental dynamics caused by other teammates. If the agent identifies the current teamwork situation in an online fashion, it can choose actions accordingly to ensure effective coordination. In this way, we introduce a multimodal representation learning framework (Suzuki et al., 2016; Yin et al., 2017). It automatically encodes the core knowledge about the teamwork situations into a latent probabilistic variable. We show that without any prior knowledge, after learning from the interactive experience with given teammates, the latent variable is sufficiently descriptive to provide information about how to coordinate with new teammates' behaviors. To overcome partial observability, we propose an information-based proxy encoder to implicitly infer the learned variables from local observations. Then, the autonomous agent adapts to new teammates' behaviors dynamically and quickly by conditioning its policy on the inferred variables.

Instead of limiting teammates into several predefined and fixed types, ODITS considers a mechanism of how an agent should adapt to teammates' behaviors online. It automatically learns continuous representations of teammates' behaviors to infer how to coordinate with current teammates' actions effectively. Without domain knowledge on current environments, it enables effective ad hoc teamwork performance and fast adaptation to varying teammates, which the agent might not thoroughly observe under partial observability. In our experimental evaluation, by interacting with a small set of given teammates, the trained agents could robustly collaborate with diverse new teammates. Compared with various type-based baselines, ODITS reveals superior ad hoc teamwork performance. Moreover, our ablations show both the necessity of learning latent variables of teamwork situations and inferring the proxy representations of learned variables.

## 2 RELATED WORKS

**Ad Hoc Teamwork.** The core challenge of achieving cooperative ad hoc teamwork is to develop an adaptive policy robust to various unknown teammates' behaviors (Stone et al., 2010). Existing type-based approaches try to predefine types of teammates and choose policies accordingly to cooperate with unknown teammates (Chen et al., 2020; Ravula et al., 2019; Durugkar et al., 2020; Mirsky et al., 2020; Barrett and Stone, 2015) . Specifically, PLASTIC (Barrett and Stone, 2015) infers types of teammates by computing Bayesian posteriors over all types. ConvCPD (Ravula et al., 2019) extends this work by introducing a mechanism to detect the change point of the current teammate's type. AATEAM (Chen et al., 2020) proposes an attention-based architecture to infer types in real time by extracting the temporal correlations from the state history. The drawback of these approaches is that finite and fixed types might not cover all possible situations in complex environments. One recent work avoids predefining teammates' types by leveraging graph neural networks (GNNs) to estimate the joint action value of an ad hoc team (Rahman et al., 2021). However, this work requires all teammates' observations as input, which might not always be available in the real world.

**Agent Modeling.** By modeling teammates' behaviors, approaches of agent modeling aims to provide auxiliary information, such as teammates' goals or future actions, for decision-making (He et al., 2016; Albrecht and Stone, 2018). For example, MeLIBA conditions the ad hoc agent's policy on a belief over teammates, which is updated following the Bayesian rule (Zintgraf et al., 2021). However, existing agent models require the full observations of teammates as input (Raileanu et al.; Grover et al.; Tacchetti et al., 2019). If the agent cannot always observe other teammates' information (e.g. observations and actions), those approaches would fail to give an accurate prediction about teammates' information. A recent work proposes to use VAE to learn fixed-policy opponents under partial observability. However, it does not generalize to the ad hoc setting where the teammates can be non-stationary (Papoudakis and Albrecht, 2020). There are also some works study how to generate diverge agent policies, which benefits the training of ad hoc agent (Canaan et al., 2019).

**Multi-agent Reinforcement Learning (MARL).** Cooperative MARL(Foerster et al., 2017) with centralized training and decentralized execution (Oliehoek et al., 2008) (CTDE) is relevant to this work. Related approaches (Sunehag et al., 2018; Rashid et al., 2018) utilize value function factorization to overcome the limitations of both joint and independent learning paradigms simultaneously. However, these algorithms assume that the developed team is fixed and closed. The team configuration (e.g., team size, team formation, and goals) is

unchanged, and agents will not meet other agents without pre-coordination. Several extended works improve the generalization ability for complex team configurations by leveraging other insights, like learning dynamic roles (Wang et al., 2021; 2020), randomized entity-wise factorization (Iqbal et al., 2020), and training regime based on game-theoretic principle (Muller et al., 2020). However, intrinsically, these approaches usually focus on co-training a group of highly-coupled agents instead of an autonomous agent that can adapt to non-stationary teammates.

## 3 BACKGROUND

**Problem Formalization.** Our aim is to develop a *single* autonomous agent, which we refer to as the *ad hoc agent*, that can effectively cooperate with various teammates under partial observability without pre-coordination, such as joint-training. While we focus on training a single agent in this work, similar approaches can be applied to construct an ad hoc team.

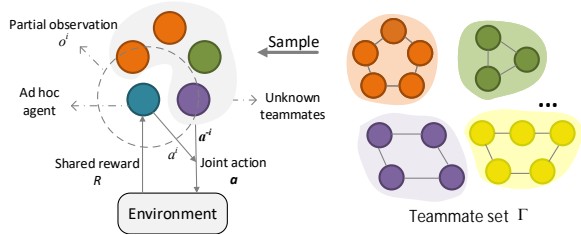

Figure 1: **Visualization of the Dec-POMDP with an addtional teammate set $\Gamma$.**

To evaluate the ad hoc agent's ability to cooperate with unknown teammates, we formally define the problem as a decentralized Partially observable Markov Decision Process (Dec-POMDP) (Oliehoek et al., 2008) with an additional assumption about the set of teammates' possible policies $\Gamma$.

It can be represented as a tuple $\langle N, \mathcal{S}, \mathcal{A}, \mathcal{O}, \mathcal{T}, P, R, O, \Gamma \rangle$, where $N$ denotes the number of agents required by the task, $s \in \mathcal{S}$ denotes the global state of the environment. The joint action $\boldsymbol{a} \in \mathcal{A}^N$ is formed by all agent's independent actions $a^i \in \mathcal{A}$, where $i$ is the index of the agent. In the environment, each agent only has access to its partial observation $o^i \in \mathcal{O}$ drawn according to the observation function $O(s, i)$, and it has an observation-action history $\tau^i \in \mathcal{T} \equiv (\mathcal{O} \times \mathcal{A})^*$. $P(s', |s, \boldsymbol{a})$ denotes the probability that taking joint action $\boldsymbol{a}$ in state $s$ results in a transition to state $s'$. $R(s, \boldsymbol{a})$ is the reward function that maps a state $s$ and a joint action $\boldsymbol{a}$ to a team reward $r \in \mathbb{R}$. $\Gamma$ represents a pool of various policies, which can be pretrained or predefined to exhibit cooperative behaviors. Without loss of generality, we denote by $\pi_i$ the policy of the ad hoc agent and by $\boldsymbol{\pi}_{-i}$ the joint policy of all other agents.

Fig.1 shows the detailed schematics of this problem. Note that the orginal group size of teammate can be arbitrary. In a Dec-POMDP with an additional teammate set $\Gamma$, the objective of the ad hoc agent is to maximize the expected team return when it teams up with $N - 1$ arbitrary teammates sampled from $\Gamma$, though it has no prior knowledge about those teammates. Therefore, the ad hoc agent's optimal policy $\pi_i^*$ is required to maximize the joint action value $Q^{\pi_i}(s, a^i, \boldsymbol{a}^{-i})$, which indicates the expected accumulative team reward over different ad hoc teamwork:

$$Q^{\pi_i}(s, a^i, \boldsymbol{a}^{-i}) = \mathbb{E}_{a^i_{t=1:+\infty} \sim \pi_i, \boldsymbol{a}^{-i}_{t=1:+\infty} \sim \boldsymbol{\pi}_{-i}, \boldsymbol{\pi}_{-i} \sim \Gamma} \left[ \sum_{t=0}^{+\infty} \gamma^t r_t \Big| s_0 = s, \boldsymbol{a}_0 = \boldsymbol{a}, P \right] \tag{1}$$

$$Q^{\pi_i^*}(s, a^i, \boldsymbol{a}^{-i}) \geq Q^{\pi_i}(s, a^i, \boldsymbol{a}^{-i}), \forall \pi_i, s, a_i, \boldsymbol{a}^{-i} \tag{2}$$

**Marginal Utility** is defined to measure the contribution of an ad hoc agent to the whole team utility (Genter and Stone, 2011). It represents the increase (or decrease) in a team's utility when an ad hoc agent is added to the team. Given teammates' actions $\boldsymbol{a}^{-i}$, there is a relationship between the marginal utility and the team utility

(denoted by the joint action value) as follow:

$$\arg\max_{a^i} u^i(s, a^i, \boldsymbol{a}^{-i}) = \arg\max_{a^i} Q^{\pi_i}(s, a^i, \boldsymbol{a}^{-i}) \tag{3}$$

where $u^i(s, a^i, \boldsymbol{a}^{-i})$ denotes the marginal utility when the ad hoc agent chooses the action $a^i$ under the state $s$. Note that the marginal utility is not necessarily equal to the Q-value (Sunehag et al., 2018). The ad hoc agent chooses the action which maximizes the marginal utility to ensure the maximal team utility.

# 4 ODITS LEARNING FRAMEWORK

Our approach addresses the ad hoc teamwork problem with a novel probabilistic framework ODITS. In this section, we first introduce the overall architecture of this framework and then present a detailed description of all modules in this framework.

## 4.1 OVERVIEW

ODITS aims to estimate the ad hoc agent's marginal utility for choosing corresponding actions to maximize the team utility. To achieve the adaptive policy to unknown teammates, we model the marginal utility as a conditional function on the inferred latent variable, which implicitly represents the current teamwork situation. ODITS jointly optimizes the marginal utility function and the latent variable by two learning objectives in an end-to-end fashion. Fig.2 shows the detailed schematics of ODITS. It splits the team into two parts: *teammates* and the *ad hoc agent*.

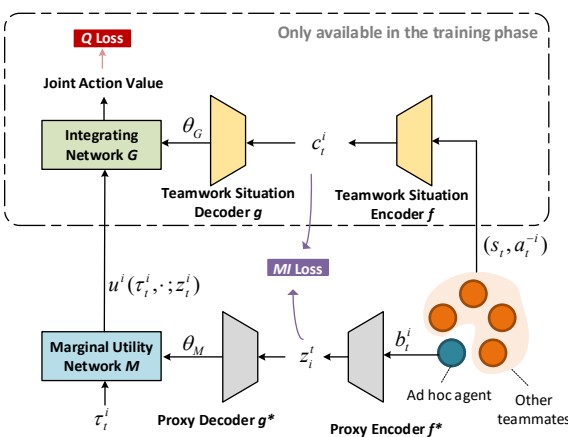

Figure 2: **Schematics of ODITS.**

First, we regard other teammates as a part of environmental dynamics perceived by the ad hoc agent. Since different combinations of teammates lead to diverse and complex dynamics, we expect to learn a latent variable to describe the core information of teammates' behaviors implicitly. To do this, we introduce a **teamwork situation encoder** $f$ to learn the variable. Then, a loss function (**Q loss**), an **integrating network** $G$ and a **teamwork situation decoder** $g$ are jointly proposed to regularize the information embedded in the learned variable $c_t^i$.

For the ad hoc agent, we expect to condition its policy on the learned variable $c_t^i$. However, the partial observability impedes the direct access to $c_t^i$. Thus, we introduce a **proxy encoder** $f^*$ to infer a proxy representation $z_t^i$ of $c_t^i$ from local observations. We force $z_t^i$ to be informationally consistent with $c_t^i$ by an information-based loss function (**MI loss**). Then, we train a **marginal utility network** $M$ to estimate the ad hoc agent's conditional marginal utility $\hat{u}^i(\tau_t^i, a_t^i; z_t^i) \approx u^i(s_t, a_t^i, \boldsymbol{a}_t^{-i})$. For conditional behavior, a part of parameters of $M$ are generated by the **proxy decoder** $g^*$.

Similar to the CTDE scenario (Oliehoek et al., 2008), we relax the partial observability in the training phase. ODITS is granted access to the global state $s_t$ and other teammates' actions $\boldsymbol{a}_t^{-i}$ during training. During execution, $G, f, g$ is removed; the ad hoc agent chooses the action which maximizes the conditional marginal utility function $\hat{u}^i(\tau_t^i, a_t^i; z_t^i)$.

## 4.2 LEARNING TO REPRESENT TEAMWORK SITUATIONS

For adaptive behaviors, we expect to condition the ad hoc agent's policy on other teammates. However, unknown teammates show complex behaviors. Directly conditioning the policy on them might lead to a volatile policy. To address this issue, we aim to embed the teammates' information into a compact but descriptive representation. To model the concept clearly, we formally define teamwork situation:

**Definition 1 (teamwork situation)** *At each time step t, the ad hoc agent is in the teamwork situation $c_t^i \in \mathcal{C}$, which is the current underlying teamwork state yielded by the environment state $s_t$ and other teammates' actions $\boldsymbol{a}_t^{-i}$. It reflects the **high-level semantics** about the teammates' behaviors.*

Though different teammates generate diverse state-action trajectories, we assume that they can cause similar teamwork situations at some time, and the ad hoc agent's action would affect their transitions. When the current teamwork situation is identified, the ad hoc agent can choose the action accordingly to ensure online adaptation.

**Teamwork Situation Encoder $f$.** To model the uncertainty of unknown teammates, we encode teamwork situations in a stochastic embedding space $\mathcal{C}$. Thus, any teamwork situation can be represented as a latent probabilistic variable $c^i$ that is drawn from a multivariate Gaussian distribution $\mathcal{N}(\mu_{c^i}, \sigma_{c^i})$. To enable the dependency revealed in the definition, we use a trainable neural network $f$ to learn the parameters of the Gaussian distribution of $c^i$:

$$(\mu_{c^i}, \sigma_{c^i}) = f(s, \boldsymbol{a}^{-i}; \theta_f), c^i \sim \mathcal{N}(\mu_{c^i}, \sigma_{c^i}) \tag{4}$$

where $\theta_f$ are parameters of $f$.

**Regularizing Information Embedded in $c^i$.** We introduce a set of modules to jointly force $c^i$ to be sufficiently descriptive for reflecting the current teamwork situation. If $c_t^i$ is able to capture the core knowledge about other teammates' current behaviors, we can predict the joint action value $Q^{\pi_i}(s_t, a_t^i, \boldsymbol{a}_t^{-i})$ according to $c_t^i$ and the ad hoc agent's marginal utility $u_t^i$ [1]. Thus, we propose an integrating network $G$ for generating the joint action value's estimation $G(u_t^i, c_t^i) \approx Q^{\pi_i}(s_t, a_t^i, \boldsymbol{a}_t^{-i})$. We adopt a modified asynchronous Q-learning's loss function (Q-loss) (Mnih et al., 2016) as the optimization objective:

$$\mathcal{L}_Q = \mathbb{E}_{(u_t^i, c_t^i, r_t) \sim \mathcal{D}} \left[ [(r_t + \gamma \max_{a_{t+1}^i} \bar{G}(u_{t+1}^i, c_{t+1}^i) - G(u_t^i, c_t^i)]^2 \right] \tag{5}$$

where $\bar{G}$ is a periodically updated target network. The expectation is estimated with uniform samples from the replay buffer $\mathcal{D}$, which saves the interactive experience with training teammates.

**Integrating Network $G$.** One simple approach for integrating $c^i$ with $u^i$ is to formulate $G$ as an MLP that maps their concatenation into the joint value estimation. We instead propose to map $c^i$ into the parameters of $G$ by a *hypernetwork* (Ha et al., 2016), which we refer to as the **teamwork situation decoder** $g$. Then, $G$ maps the ad hoc agent's utility $u^i$ into the value estimation. This alternative design changes the procedure of information integration. The decoder provides *multiplicative integration* to aggregate information. By contrast, the concatenation-based operation only provides *additive integration*, leading to a poor information integration ability (see Supplementary). We also empirically show that multiplicative integration stabilizes the training procedure and improves teamwork performance. In addition, we expect that there is a **monotonicity** modeling the relationship between $G$ and the marginal utility $u_t^i : \frac{\partial G}{\partial u_t^i} \geq 0$. Given any $c_t^i$, the increase of the ad hoc agent's marginal utility results in the improved joint action value. To achieve this property, we force $\theta_G \geq 0$.

### 4.3 LEARNING CONDITIONAL MARGINAL UTILITY FUNCTION UNDER PARTIAL OBSERVABILITY

Apparently, the marginal utility of the ad hoc agent depends on other teammates' behaviors. Distinct behaviors result in different marginal utilities. Here, we formalize the marginal utility network $M$ as a deep recurrent Q network (DRQN) (Hausknecht and Stone, 2015) parameterized by $\theta_M$. To enable adaptation, we force the final layers' parameters of $M$ to condition on the learned variable $c_t^i$.

**Proxy Encoder $f^*$.** Because of partial observability, the teamwork situation encoder $f$ is not available during execution. Thus, we introduce a proxy encoder $f^*$ to estimate $c_t^i$ from the local transition data $b_t^i = (o_t^i, r_{t-1}, a_{t-1}^i, o_{t-1}^i)$. We assume that $b_t^i$ can partly reflect the current teamwork situation since the transition implicitly indicates the underlying dynamics, which is primarily influenced by other teammates' behaviors. We denote the estimation of $c_t^i$ as $z_t^i$. Then, $z_t^i$ would be fed into a *proxy decoder* $g^*(z_t^i; \theta_{g^*})$ parameterized by $\theta_{g^*}$ to generate the parameters $\theta_M$ of $M$, enabling the marginal utility function to condition on the proxy representation $z_t^i$. Similar to $c^i$, we encode $z^i$ into a stochastic embedding space:

$$(\mu_{z^i}, \sigma_{z^i}) = f^*(b^i; \theta_{f^*}), z^i \sim \mathcal{N}(\mu_{z^i}, \sigma_{z^i}) \tag{6}$$

where $\theta_{f^*}$ are parameters of $f^*$.

---

[1] $u_t^i$ is a shorthand notation of $u^i(\tau_t^i, a_t^i; z_t^i)$

**Regularizing Information Embedded in $z^i$.** To make $z_t^i$ identifiable, we expect $z_t^i$ to be informatively consistent with $c_t^i$. Thus, we introduce an information-based loss function $\mathcal{L}_{MI}$ here to maximize the conditional mutual information $I(z_t^i; c_t^i | b_t^i)$ between the proxy variables and the true variables. However, estimating and maximizing mutual information is often infeasible. We introduce a variational distribution $q_\xi(z_t^i | c_t^i, b_t^i)$ parameterized by $\xi$ to derive a tractable lower bound for the mutual information (Alemi et al., 2017):

$$I(z_t^i; c_t^i | b_t^i) \geq \mathbb{E}_{z_t^i, c_t^i, b_t^i} \left[ \log \frac{q_\xi(z_t^i | c_t^i, b_t^i)}{p(z_t^i | b_t^i)} \right] \tag{7}$$

where $p(z_t^i | b_t^i)$ is the Gaussian distribution $\mathcal{N}(\mu_{z^i}, \sigma_{z^i})$. This lower bound can be rewritten as a loss function to be minimized:

$$\mathcal{L}_{MI}(\theta_{f^*}, \xi) == \mathbb{E}_{(b_t^i, s^t, \boldsymbol{a}_t^{-i}) \sim \mathcal{D}}[D_{KL}[p(z_t^i | b_t^i) || q_\xi(z_t^i | c_t^i, b_t^i)]] \tag{8}$$

where $\mathcal{D}$ is the replay buffer, $D_{KL}[\cdot || \cdot]$ is the KL divergence operator. The detailed derivation can be found in Supplementary.

## 4.4 Overall Optimization Objective

---
**Algorithm 1** ODITS Training
---
**Require:** Batch of training teammates' behavioral policies $\{\boldsymbol{\pi}_j^{-i}\}_{j=1,2,\cdots,J}^{tr}$ ; learning rate $\alpha$; scaling factor $\lambda$.
1: initialize the replay buffer $\mathcal{D}$
2: **while** not done **do**
3:      Sample the teammates' policies $\boldsymbol{\pi}_j^{-i}$ from $\{\boldsymbol{\pi}_j^{-i}\}_{j=1,2,\cdots,J}^{tr}$
4:      **for** $k = 1, \cdots, K$ **do**
5:          Sample data $D_k = \{(s_t, a_t^i, \boldsymbol{a}_t^{-i}, r_t)\}_{t=1,\cdots,T}$ using the ad hoc agent's policy $\pi^i$ and $\boldsymbol{\pi}^{-i}$
6:          Add $D_k$ into $\mathcal{D}$
7:      **for** steps in training steps **do**
8:          Sample one trajectory $D \sim \mathcal{D}$
9:          **for** $t = 1, \cdots, T-1$ **do**
10:              Compute $(\mu_{c_t^i}, \sigma_{c_t^i}) = f(s_t, \boldsymbol{a}_t^{-i})$ and sample $c_t^i \sim \mathcal{N}(\mu_{c_t^i}, \sigma_{c_t^i})$
11:              Compute $(\mu_{z_t^i}, \sigma_{z_t^i}) = f^*(b_t^i)$ and sample $z_t^i \sim \mathcal{N}(\mu_{z_t^i}, \sigma_{z_t^i})$
12:              Compute $u_t^i(\tau_t^i, a_t^i; z_t^i)$ and $G(u_t^i; c_t^i)$
13:              Compute $\mathcal{L}_Q, \mathcal{L}_{MI}$
14:              $\theta \leftarrow \theta + \alpha \nabla_\theta(\mathcal{L}_Q)$
15:              $\theta_{f^*} \leftarrow \theta_{f^*} + \lambda \cdot \alpha \nabla_{\theta_{f^*}}(\mathcal{L}_{MI})$
16:              $\xi \leftarrow \xi + \lambda \cdot \alpha \nabla_\xi(\mathcal{L}_{MI})$
---

---
**Algorithm 2** ODITS Testing
---
**Require:** Testing teammates' behavioral policies $\boldsymbol{\pi}^{-i}$.
1: **for** $t = 1, \cdots, T$ **do**
2:      Generate teaammtes' actions $\boldsymbol{a}_t^{-i} \sim \boldsymbol{\pi}^{-i}$
3:      Compute $(\mu_{z_t^i}, \sigma_{z_t^i}) = f^*(b_t^i)$ and sample $z_t^i \sim \mathcal{N}(\mu_{z_t^i}, \sigma_{z_t^i})$
4:      Do the action $a_t^i$ that maximizes $u^i(\tau_t^i, a_t^i; z_t^i)$
---

To the end, the overall objective becomes:

$$\mathcal{L}(\theta) = \mathcal{L}_Q(\theta) + \lambda \mathcal{L}_{MI}(\theta_{f^*}, \xi) \tag{9}$$

where $\theta = (\theta_f, \theta_g, \theta_M, \theta_{f^*}, \theta_p, \xi)$, $\lambda$ is the scaling facor.

During the training phase, the ad hoc agent interacts with different training teammates for collecting transition data into the replay buffer $\mathcal{D}$. Then, samples from $\mathcal{D}$ are fed into the framework for updating all parameters

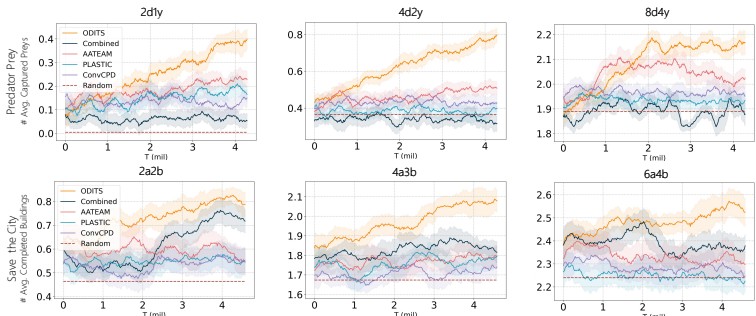

Figure 3: Performance comparison across various scenarios for Predator Prey (**top panel**) and Save the City (**bottom panel**).

by gradients induced by the overall loss. During execution, the ad hoc agent conditions its behavior on the inferred teamwork situations by choosing actions to maximize the conditional utility function $u^i(\tau_t^i, a_t^i; z_t^i)$. We summarize our training procedure and testing procedure in Algorithm 1 and Algorithm 2.

## 5 EXPERIMENTS

We now empirically evaluate ODITS on various new and existing domains. All experiments in this paper are carried out 4 different random seeds, and results are shown with a 95% confidence interval over the standard deviation. In the following description, we refer to the teammates that interact with the ad hoc agent during the training phase as the training teammates, and refer to those teammates with unknown policies as testing teammates. And the "teammate types" correspond to the policy types of teammates. All experimental results illustrate the average teamwork performance when the ad hoc agent cooperates with different testing teammates. Additional experiments, further experiment details and implementation details of all models can be found at Supplementary.

### 5.1 PREDATOR PREY

**Configurations.** In this environment, $m$ homogenous predators try to capture $n$ randomly-moving preys in a $7 \times 7$ grid world. Each predator has six actions: the moving actions in four directions, the capturing action, and waiting at a cell. Due to partial observability, each predator can only access the environmental information within two cells nearby. Besides, there are two obstacles at random locations. Episodes are 40 steps long. The predators get a team reward of 500 if two or more predators are capturing the same prey at the same time, and they are penalized for -10 if only one of them tries to capture a prey. Here, we adopt three different settings to verify the ad hoc agent's ability to cooperate with different number of teammates. They are 2 predators and 1 preys (2d1y), 4 predators and 2 preys (4d2y) and 8 predators and 4 preys (8d4y), respectively.

We compare our methods with three type-based baselines: AATEAM (Chen et al., 2020), ConvCPD (Ravula et al., 2019), PLASTIC (Barrett and Stone, 2015). Note that these approaches assume that the ad hoc agent has full visibility on the environment. To apply them on partially observed settings, we replace the full state information used in them with partial observations of the ad hoc agent. Furthermore, we also compare two other strategies: (i) Random: The ad hoc agent chooses actions randomly. (ii) Combined: The ad hoc agent utilizes a DQN algorithm to learn a single policy using the data collected from all possible teammates. This intuitive baseline provides the view of treating the problem as a vanilla single-agent learning problem, where the agent ignores the differences between its teammates.

Before training all algorithms, we first require a teammate set that consists of various behavioral policies of teammates. Instead of crafting several teammates' cooperative policies by hand, we expect to train a set of distinct policies automatically. Therefore, we first utilize 5 different MARL algorithms (e.g. VDN (Sunehag et al., 2018) and QMIX (Rashid et al., 2018)) to develop several teams of agents. To ensure diversity, we use different random seeds for each algorithm and save the corresponding models at 3 different checkpoints (3 million steps, 4 million

steps, and 5 million steps). Then, we manually select 15 different policies showing distinct policy representations (Grover et al.) from all developed models. Finally, we randomly sampled 8 policies as the training set and the other 7 policies as the testing set. During training, we define 8 teammate types that correspond to 8 policies in the training set for the type-based approaches. Then, algorithms would develop their models according to the interactive experience with training teammates. For all algorithms, agents are trained for 4.5 million time steps. The number of captured preys when the ad hoc agent cooperates with testing teammates throughout training is reported. See Supplementary for further settings.

**Results.** The top panel of Fig. 3 reports the results across 3 scenarios. We first observe that ODITS achieves superior results on the number of captured preys across a varying number of teammates, verifying its effectiveness. ODITS also tends to give more consistent results than other methods across different difficulties. The other 3 type-based baselines and ODITS show better results than random and combined policies, indicating that they can indeed lead to adaptive behaviors to different teammates. Furthermore, the random strategy captures some preys on `4d2y` and `8d4y`, but completely fails on `2d1y`. This indicates that without the cooperative behaviors of the ad hoc agent, other teammates can also coordinate with each other to achieve the given goal. The combined policy shows worse results than the random policy on two scenarios (`4d2y` and `8d4y`). This might be because the combined policy show behaviors that conflict with other teammates. With the number of teammates increasing, the increasing effects of conflicts lead to serious miscoordination.

## 5.2 SAVE THE CITY

**Configurations.** This is a grid world resource allocation task presented in (Iqbal et al., 2020). In this task, there are 3 distinct types of agents, and their goal is to complete the construction of all buildings on the map while preventing them from burning down. Each agent has 8 actions: stay in place, move to the next cell in one of the four cardinal directions, put out the fire, and build. We set the agents to get a team reward of 100 if they have completed a building and be penalized for -500 when one building is burned down. Agent types include firefighters (20x speedup over the base rate in reducing fires), builders (20x speedup in the building), or generalists (5x speedup in both as well 2x speedup in moving ). Buildings also consist of two varieties: fast-burning and slow-burning, where the fast-burning buildings burn four times faster. In our experiments, each agent can only access the environmental information within four cells nearby. We adopt three different scenarios here to verify all methods. They are 2 agents and 1 buildings (`2a2b`), 4 agents and 3 buildings (`4a3b`), 6 agents and 4 buildings (`6a4b`).

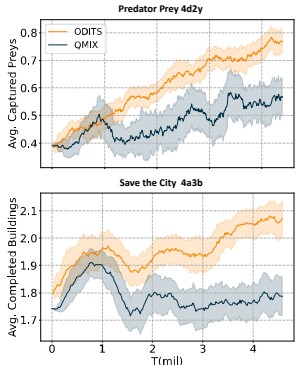

Similar to training settings in Predator Prey, we select 15 distinct behavioral policies for the teammate set and randomly partition them into 8 training policies and 7 testing policies. For all algorithms, agents are trained for 4.5 million time steps. The number of completed buildings when the ad hoc agent cooperates with testings teammates throughout training is reported. See Supplementary for further settings.

Figure 4: ODITS V.S. QMIX.

**Results.** The bottom panel of Fig. 3 reports the results across 3 scenarios. We first observe that ODITS outperforms other baselines, verifying its effectiveness. Since the setting force all agents in the environment to be heterogeneous, the results also underpin the robustness of ODITS. Interestingly, we find that the combined policy reveals better performance than other type-based approaches. This result is not consistent with that in Predator Prey. Our intuition is that the requirement of cooperative behaviors in Save the City is less than that in Predator Prey. Actually, one agent in Save the City can complete buildings individually without the strong necessity of cooperating with other teammates' behaviors. By contrast, one predator cannot capture prey by itself. As a result, the combined policy learns a universal and effective policy by learning from more interactive experience with different teammates, while type-based approaches fail because developing distinct cooperative behaviors leads to the instability of the ad hoc agent. This hypothesis is also empirically demonstrated in our ablations.

## 5.3 COMPARISON WITH MARL

In order to compare the performance of ODITS and MARL, we implement the commonly used algorithm QMIX (Rashid et al., 2018) as our baseline. Similar to the training procedure of ODITS, we fix one agent and train it with teammates randomly sampled from a pool consisting of 8 policies. The gradients for updating the

teammates' policies are blocked but the mixing network is updating as in the original implementation of QMIX. Figure 4 shows the comparison of ODITS and QMIX on Predator Prey `4d2y` and Save the City `4a3b`. In both environments, QMIX performs significantly worse than ODITS. This is not quite surprising because MARL algorithms usually assume that all the teammates are fixed. Therefore, although trained with multiple teammates, the agent under the QMIX framework does not learn to cooperate with an impromptu team.

### 5.4 ABLATIONS.

We perform several ablations on the Predator Prey `4d2y` and Save the City `4a3b` to try and determine the importance of each component of ODITS.

**Adaptive Behaviors.** We first consider removing the information-based loss $\mathcal{L}_{MI}$ from the overall learning objective ( denoted as **w/o info.**), Fig. 5 shows that without $\mathcal{L}_{MI}$ regularizing the information embedded in $z_t^i$, ODITS induces worse teamwork performance. This indicates that improving the mutual information between the proxy variable and the true variable indeed results in better representations of teamwork situations. We next consider how the inferred variables of teamwork situations affect the ad hoc agent's adaptive behaviors. We remove the proxy encoder and set $z_t^i$ as a fixed and randomly generated vector ( denoted as **w/o infer.**). As shown in

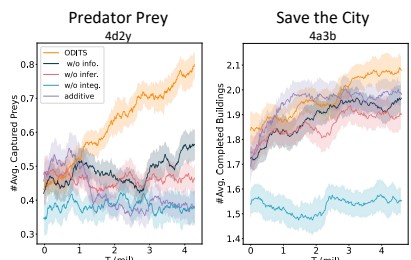

Figure 5: Ablations for different components.

Fig. 5, conditioning on a random signal leads to a further drop in performance, indicating that irrelevant signals cannot promote the ad hoc agent to develop adaptive policies.

**Integrating Mechanism.** We remove the teamwork situation encoder as well as $\mathcal{L}_{MI}$ from the framework and feed a vector filled with 1 into the teamwork situation decoder (labeled with **w/o integ.**). This setting enables ODITS not to integrating the ad hoc agent's marginal utility with the information of teammates' behaviors. Compared with **w/o info.**, it brings a larger drop in teamwork performance. One intuition is that predicting the joint-action value plays an essential role in estimating the marginal utility. Suppose the integrating network has no information on other teammates' behaviors. In that case, it cannot accurately predict the joint-action value, resulting in instability in marginal utility estimation. Despite the empirical evidence supporting this argument, however, it would be interesting to develop further theoretical insights into this training regime in future work. We finally consider the additive integration mechanism mentioned in section 4.2 (labeled with **additve**). We observe that despite additive integration shows an excellent performance in Save the City, it suffers from poor performance in Predator Prey, indicating that multiplicative integration provides a more stable and effective ability to integrate information from teammates and the ad hoc agent. Interestingly, we also find that most ablations get worse results in Predator Prey than those in Save the city. We believe that the different levels of cooperative requirement in two environments result in this phenomenon. The prey is captured when two nearby predators are simultaneously capturing them. By contrast, the burning building can be constructed by an individual agent. Therefore, removing mechanisms that promote the cooperative behaviors leads to the worse performance in Predator Prey.

## 6 CONCLUSIONS

This paper proposes a novel adaptive reinforcement learning algorithm called ODITS to address the challenging ad hoc teamwork problem. Without the need to predefine types of teammates, ODITS automatically learns compact but descriptive variables to infer how to coordinate with previously unknown teammates' behaviors. To overcome partial observability, we introduce an information based regularizer to estimate proxy representations of learned variables from local observations. Experimental results show that ODITS obtains superior performance compared to various baselines on several complex ad hoc teamwork benchmarks.

ACKNOWLEDGMENTS

This research was supported by the National Research Foundation, Singapore under its AI Singapore Programme (AISG Award No: AISG-RP-2019-0013), National Satellite of Excellence in Trustworthy Software Systems (Award No: NSOETSS2019-01), and NTU.

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

# A  APPENDIX

## A.1  MATHEMATICAL DERIVATION

### A.1.1  ADDITIVE INTEGRATION VS. MULTIPLICATIVE INTEGRATION IN INTEGRATING NETWORK.

Concretely, let us consider a simple example that $G$ is a one-layer network. The information aggregation of additive integration (abbreviated as $G_{add}$) could be written as follow:

$$G_{add} = \mathcal{F}(\boldsymbol{W_u}\boldsymbol{u^i} + \boldsymbol{W_c}\boldsymbol{c^i} + \boldsymbol{b}) \tag{10}$$

where $\boldsymbol{W_u}$ and $\boldsymbol{W_c}$ are the weight vectors of $u^i$ and $c^i$, respectively; $\boldsymbol{b}$ is the bias vector; $\mathcal{F}$ is the corresponding activation function.

By comparison, the multiplicative integration (abbreviated as $G_{mul}$) could be written as follow:

$$G_{mul} = \mathcal{F}(\boldsymbol{W_c}(\boldsymbol{c^i})\boldsymbol{W_u}\boldsymbol{u^i} + \boldsymbol{b}(\boldsymbol{c^i})) \tag{11}$$

where $\boldsymbol{W}(\boldsymbol{c^i})$ and $\boldsymbol{b}(\boldsymbol{c^i})$ are the function of $\boldsymbol{c^i}$ for generating weight vectors.

Compared with $G_{add}$, $G_{mul}$ changes the integrating network from first order to second order, while introducing no extra parameters. The information of teamwork situation directly scales the information of the ad hoc agent through the term $\boldsymbol{W}(\boldsymbol{c^i})\boldsymbol{W_u}\boldsymbol{u^i}$, rather than via a more subtle bias $\boldsymbol{W_u}\boldsymbol{u^i} + \boldsymbol{W_c}\boldsymbol{c^i}$.

Further, $G_{mul}$ also brings about advantages over $G_{add}$ as it alters the integrating network's gradient properties. In the additive integration, the gradient of $\frac{\partial G_{add}}{\partial c^i}$ can be computed as follow:

$$\frac{\partial G_{add}}{\partial c^i} = \boldsymbol{W_c}\mathcal{F}' \tag{12}$$

where $\mathcal{F}' = \mathcal{F}'(\boldsymbol{W_u}\boldsymbol{u^i} + \boldsymbol{W_c}\boldsymbol{c^i} + \boldsymbol{b})$. This equation reveals that the gradient heavily depends on the matrix $\boldsymbol{W_c}$, but $\boldsymbol{W_u}$ and $\boldsymbol{u^i}$ play a limited role: they only come in the derivative of $\mathcal{F}'$ mixed with $\boldsymbol{W_u}\boldsymbol{u^i}$. By comparison, $\frac{\partial G_{add}}{\partial c^i}$ is as follow:

$$\frac{\partial G_{mul}}{\partial \boldsymbol{c^i}} = [\boldsymbol{W_u}\boldsymbol{u^i}\boldsymbol{W_c'} + \boldsymbol{b'}]\mathcal{F}' \tag{13}$$

where $\boldsymbol{W_c'} = \boldsymbol{W_c'}(\boldsymbol{c^i})$ and $\boldsymbol{b'} = \boldsymbol{b'}(\boldsymbol{c^i})$. In this equation, $\boldsymbol{W_u}\boldsymbol{u^i}$ is directly involved in the gradient computation by gating $\boldsymbol{W_c'}$, hence more capble of altering the updates of the learning procedure. This naturally leads to a better information integration.

### A.1.2  MUTUAL INFORMATION LOSS FUNCTION $\mathcal{L}_{MI}$

For forcing the proxy representation of the learned latent variable to be infomationally consistent with the true latent variable, we propose to maximize the mutual information between the proxy representations and the latent variables. In this paper, we introduce a posterior estimator and derive a tractable lower bound of the mutual information term:

$$
\begin{aligned}
I(z_t^i; c_t^i | b_t^i) &= \mathbb{E}_{z_t^i, c_t^i, b_t^i}\left[\log \frac{p(z_t^i | c_t^i, b_t^i)}{p(z_t^i | b_t^i)}\right] \\
&= \mathbb{E}_{z_t^i, c_t^i, b_t^i}\left[\log \frac{q_\xi(z_t^i | c_t^i, b_t^i)}{p(z_t^i | b_t^i)}\right] \\
&\quad + \mathbb{E}_{c_t^i, b_t^i}[D_{KL}(p(z_t^i | c_t^i, b_t^i) || q_{xi}(z_t^i | c_t^i, b_t^i))] \\
&\geq \mathbb{E}_{z_t^i, c_t^i, b_t^i}\left[\log \frac{q_\xi(z_t^i | c_t^i, b_t^i)}{p(z_t^i | b_t^i)}\right]
\end{aligned} \tag{14}
$$

where the last inequality holds because of the non-negativity of the KL divergence. Then it follows that:

$$\mathbb{E}_{z_t^i, c_t^i, b_t^i} \left[ \log \frac{q_\xi(z_t^i|c_t^i, b_t^i)}{p(z_t^i|b_t^i)} \right]$$

$$= \mathbb{E}_{z_t^i, c_t^i, b_t^i}[\log q_\xi(z_t^i|c_t^i, b_t^i)] - \mathbb{E}_{z_t^i, b_t^i}[\log p(z_t^i|b_t^i)]$$

$$= \mathbb{E}_{z_t^i, c_t^i, b_t^i}[\log q_\xi(z_t^i|c_t^i, b_t^i)] + \mathbb{E}_{b_t^i}[H(z_t^i|b_t^i)] \qquad (15)$$

$$= \mathbb{E}_{c_t^i, b_t^i} \left[ \int p(z_t^i|c_t^i, b_t^i) \log q_\xi(z_t^i|c_t^i, b_t^i) dz_t^i \right] + \mathbb{E}_{b_t^i}[H(z_t^i|b_t^i)]$$

The proxy encoder is conditioned on the transition data. Given the transitions, the distribution of the proxy representations $p(z_t^i)$ are independent from the local histories. Thus, we have

$$I(z_t^i; c_t^i|b_t^i) \geq -\mathbb{E}_{c_t^i, b_t^i}[\mathcal{CE}[p(z_t^i|c_t^i, b_t^i)||q_\xi(z_t^i|c_t^i, b_t^i)dz_t^i] + \mathbb{E}_{b_t^i}[H(z_t^i|b_t^i)] \qquad (16)$$

where In practice, we sample data from the replay buffer $\mathcal{D}$ and minimize

$$\mathcal{L}_{MI}(\theta_{f^*}, \xi) = -\mathbb{E}_{(c_t^i, b_t^i) \sim \mathcal{D}}[\mathcal{CE}[p(z_t^i|c_t^i, b_t^i)||q_\xi(z_t^i|c_t^i, b_t^i)dz_t^i] + \mathbb{E}_{b_t^i}[H(z_t^i|b_t^i)]$$

$$= \mathbb{E}_{(b_t^i, s^t, \boldsymbol{a}_t^{-i}) \sim \mathcal{D}}[D_{KL}[p(z_t^i|b_t^i)||q_\xi(z_t^i|c_t^i, b_t^i)]] \qquad (17)$$

## A.2 ARCHITECTURE, HYPERPARAMETERS, AND INFRASTRUCTURE

### A.2.1 ODITS

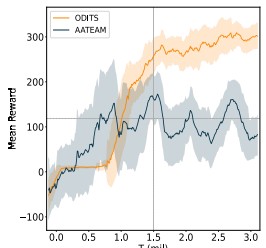

Details of the neural network architectures used by ODITS in all environments are provided in Fig. 7. We control the scale of the loss function by controlling the optimization procedure. It is conducted using RMSprop with a learning rate of $5 \times 10^{-4}$, $\alpha$ of 0.99, and with no momentum or weight decay. For the lambda value, we search over $\{1e-5, 1e-4, 5e-4, 1e-3, 5e-3, 1e-2\}$. We finally adopt $\lambda_{MI} = 1e-3$, $\lambda_{MI} = 1e-3$, and $\lambda_{MI} = 5e-4$ for Modified Coin Game, Predator Prey, and Save the City, respectively, since they induce the best performance compared with other values. For the dimension of the latent variables $z_t^i$ and $c_t^i$, we search over $\{1, 2, 3, 5, 10\}$ and finally adopt $|z| = 10$ in Save the city and $|z| = 1$ in the other environments. In addition, we set $|c| = |z|$. For exploration, we use $\epsilon$-greedy with $\epsilon$ from 1.0 to 0.05 over $50,000$ time steps and kept constant for the rest of the training. Batches of 128 episodes are sampled from the replay buffer, and all components in the framework are trained together in an end-to-end fashion.

Figure 6: performance comparison of two algorithms on the modified coin game.

### A.2.2 BASELINES

We compare ODITS with 5 baselines. For the `random` strategy, we force the ad hoc agent to choose its action at each time step randomly.

For the `combined` strategy, a three-layer DQN is trained to get the policy of the ad hoc agent. Details of DQN's architecture are illustrated in Fig. 7. We sample a type of training teammates at each training episode and collect the interacting experience data into the replay buffer. The optimization procedure and the exploration scheme are the same as those of ODITS. We set the batch size of samples as 128 and the target update interval as 200 episodes.

For the other three type-based baselines, we assume that each training policy of teammates corresponds to a predefined type of teammates. So, the number of predefined types equals to the number of training polices. We construct a set of three-layer DQN to learn the policies for all training types. Each DQN only learns the corresponding teammates' of interacting experience. Training settings for these DQNs are the same as those used in the `combined` strategy. Furthermore, to apply these baselines in partially observable environments, we replace the state information used in them with partial observations. For `PLASTIC` (Barrett and Stone, 2015),

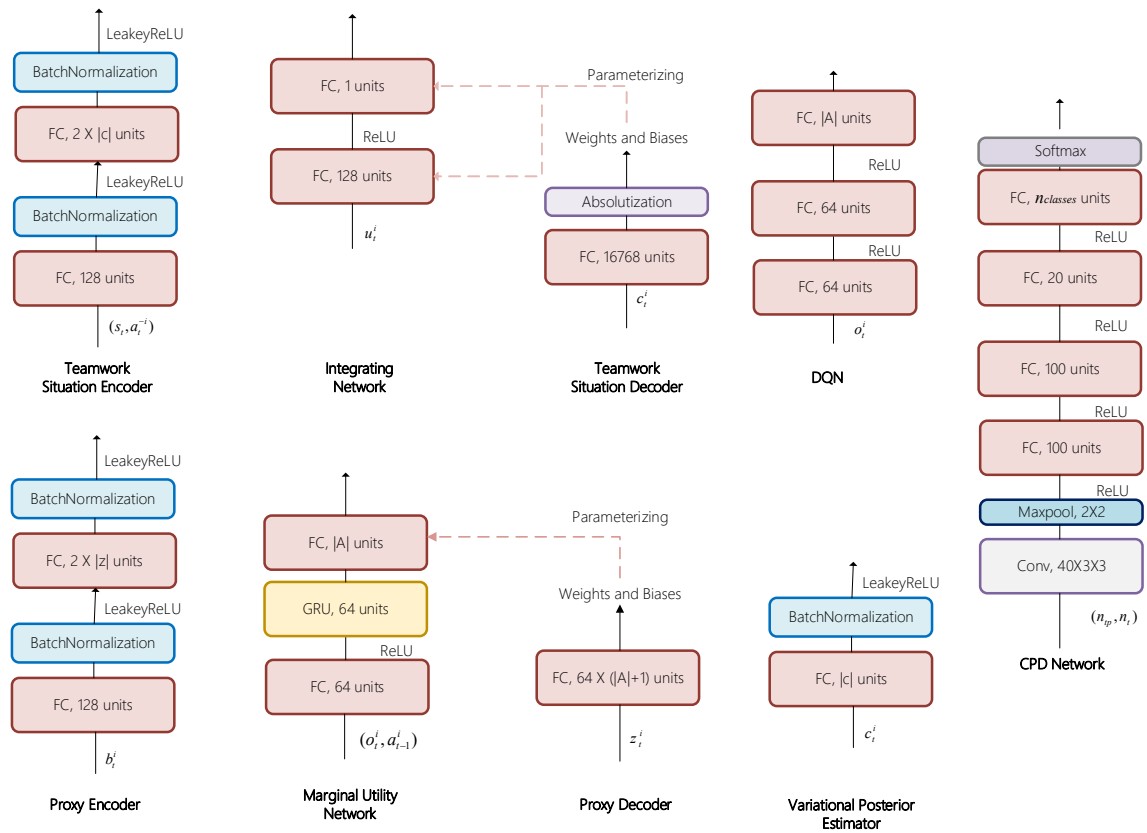

Figure 7: **Architecture details of ODITS and baselines.**

we set the parameter $\eta$ used in the UpdateBelief function as 0.2. For `ConvCPD` (Ravula et al., 2019), we follow the implementation details mentioned in the original paper. We construct the architecture of the Change Point Detection (CPD) Network as illustrated in Fig. 7, where $n_{tp}$ is the number of predefined types, $n_t$ is the number of time steps that needed be considered in ConvCPD, $n_{classes} = n_{tp} \times (n_{tp} - 1) + 1$ is the number of classess of change points. The ConCPD algorithm is trained with $n_{classes} \times 1000$ samples in which each claess has 1000 samples (batch size = 64, learning rate = 0.01, decay=0.1, optimizer=SGD). For `AATEAM` (Chen et al., 2020), we follow its proposed implementation details. The number of hidden layers for GRU is 5. The maximum length of a state sequence that attention networks process is 60. The rate of dropout is 0.1. During training, the default learning rate is 0.01, and the batch size is 32.

## A.3 MODIFIED COIN GAME

To show the difference between ODITS and type-based approaches, we introduce a simple modified coin game. The game takes place on a $7 \times 7$ map which contains 6 coins of 3 different colors (2 coins of each color). The aim of the team is to only collect any two kinds of coins (correct coins with a reward of 100) and avoid collecting the other kind of coins (false coins with a reward of -200). The policies of the teammates are predefined and illustrated in the order of colors in Fig.8 left (2 training types and 2 testing types). For example, the first training type (red $\rightarrow$ green) indicates that the policy of this teammate is to collect red and green coins, and it will collect red coins firstly. Therefore, while the correct coins of the first training type (green $\rightarrow$ red) and the second testing type (red $\rightarrow$ green) are the same, they are different policies since their paths to collect coins are apparently different. Each agent has five actions: move up, down, left, right, or pass. Once an agent steps on a coin, that

coin disappears from the cell. The game ends after 20 steps. To maximize the team return, the aim of the ad hoc agent is to infer its current teammates' desired coins and collect them as much as possible.

Here, we adopt one state-of-the art type-based approach (`AATEAM` (Chen et al., 2020)) as the baseline. Fig.6 shows the testing performance. We observe that ODITS shows superior performance and converges quickly while AATEAM shows an unstable curve. We believe this discrepance results from the key difference between our method and type-based approaches. The baseline is hard to cooperate with new types of teammates. For example, when the baseline agent interacts with the teammate of the second testing type (green → red) and observes that the teammate is collecting the green coins at the start stage, it would switch its own policy to the one corresponding to the second training type of teammate (green → blue), so it would collect green coins and blue coins (false coins) simultaneously, leading to poor teamwork performance. By contrast, ODITS can be easily generalized to the testing types of teammates. During training, ODITS learns how to cooperate with the teammate according to its current behavior instead of its types. If it observes that its teammate is collecting one kind of coins, it will collect the same kind of coins, and this knowledge is automatically embedded in $c$ and $z$.

## A.4 DETAILS OF EXPERIMENTS

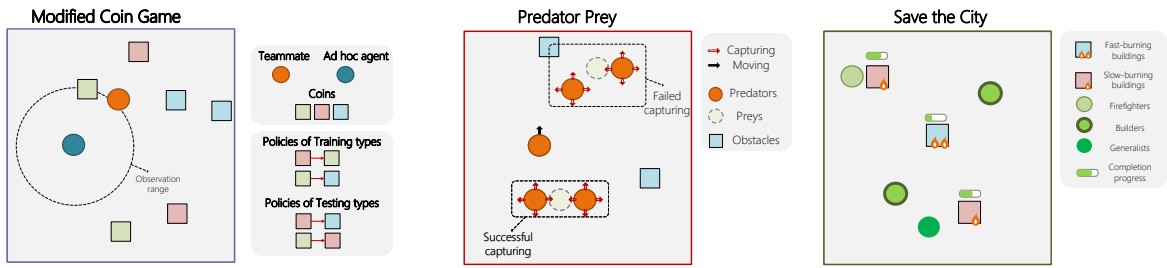

Figure 8: Illustration of Modified Coin Game (**left**), Predator Prey (**middle**) and Save the City (**right**)

**Modified Coin Game.** In this environment, the ad hoc agent needs to cooperate with its teammate to collect target coins which depends on the current teammates' type. The teammates' types are illustrated as two ordered colors of coins in Fig. 8. The teammate follows a heuristic strategy and has complete visibility of the entire environmental state. It would first move along an axis for which it has the smallest distance from the coin of its first color. If it collides with obstacles or boundaries, it would first choose random actions in 3 steps, and then choose actions to move to the target coin again. If there is no coin of its first color, it would move to the coin of its second color following the same strategy. The ad hoc agent can only access the information 3 grids nearby itself. We use one-hot coding to represent different entities in each grid and concatenate them together as a vector to construct the observation of the ad hoc agent. The game ends after 20 steps and takes place on a $7 \times 7$ grid containing 6 coins of 3 different colors (2 coins of each color). Each agent has five actions: move up, down, left, right, or pass. Once an agent steps on a coin, that coin disappears from the grid. This game requires agents to collect as many target coins as possible, which are indicated by the teammates' types. One right collected coin gives a reward of 100, while one false coin gives a reward of -200. As a result, the ad hoc agent needs to infer its teammate's goals according to teammates' behaviors and move to right coins while avoiding meeting false coins.

**Predator Prey.** In this environment, $m$ homogenous predators try to capture $n$ randomly-moving preys in a $7 \times 7$ grid world. Each predator has six actions: the moving actions in four directions, the capturing action, and waiting at a grid. Besides, there are two obstacles at random locations. Due to partial observability, each predator can only access the environmental information within two grids nearby. The information of each grid is embedded into a one-hot vector to represent different entities: obstacles, blank grids, predators, preys. Episodes are 40 steps long. The predators get a team reward of 500 if two or more predators are capturing the same prey simultaneously, and they are penalized for -10 if only one of them tries to capture a prey. Here, we adopt three different scenarios to verify the ad hoc agent's ability to cooperate with the different number of teammates. They are 2 predators and 1 preys (`2d1y`), 4 predators and 2 preys (`4d2y`) and 8 predators and 4 preys (`8d4y`), respectively.

In this environment, to simulate complex cooperative behaviors, we utilize 5 algorithms (VDN (Sunehag et al., 2018), MAVEN (Mahajan et al., 2019), ROMA (Wang et al., 2020), QMIX (Rashid et al., 2018), IQL (Tan,

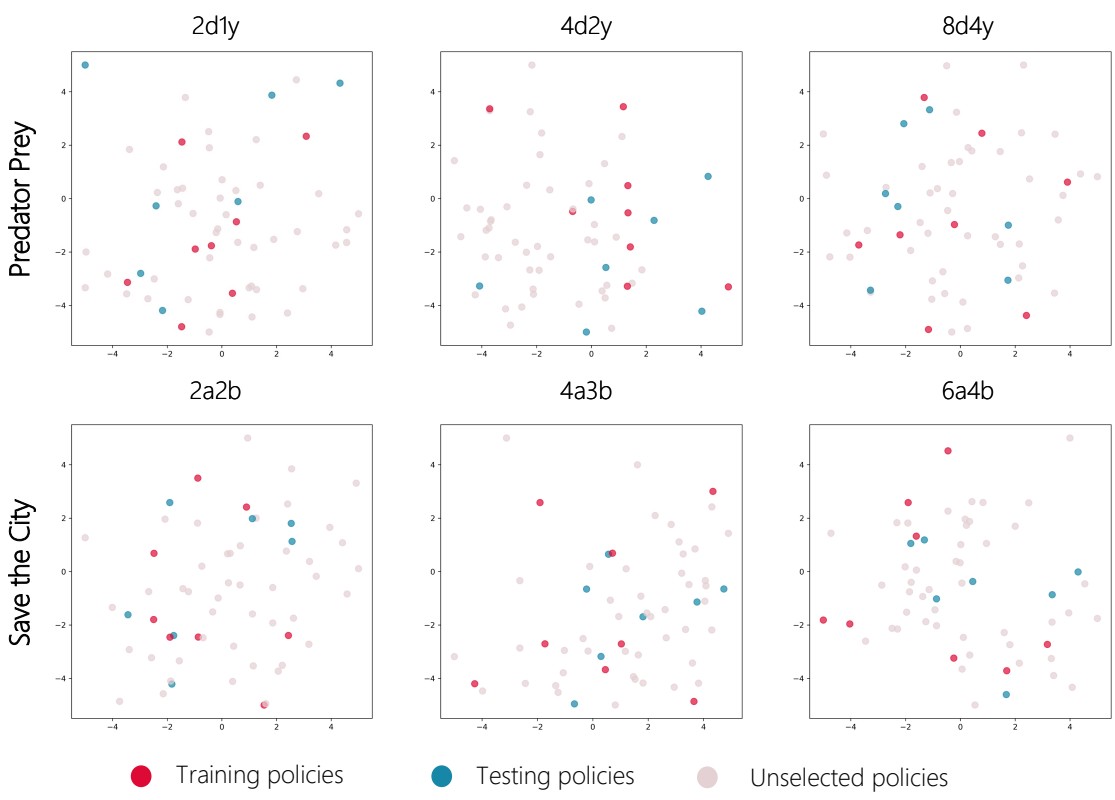

Figure 9: t-SNE plots of the learned teammates' policy representations for Predator Prey (**top panel**) and Save the City (**bottom panel**) in 6 scenarios. For each scenario, we first developed 60 candidate policies by using existing MARL open-source implementations. Then, we train a self-supervised policy encoder mentioned in (Raileanu et al., 2020) by learning information from collected behavioral trajectories $\{(o_t^i, a_t^i)\}$ of all candidates to represent their policies. Finally, we select 15 policies whose average policy embeddings are distinct from each other, and split them into 8 training policies and 7 testing policies for each scenario.

1993)) to train the candidate policies of teammates using their open-source implementations based on PyMARL (Samvelyan et al., 2019). We use 4 different random seeds for each algorithm and save the corresponding models at 3 different training steps (3 million, 4 million, and 5 million). As a result, we get 60 candidate teammates' policies for each scenario. To ensure diversity of cooperative behaviors, we visualize policy representations of all candidate models by using the open-source implementation mentioned in (Raileanu et al., 2020). This mechanism encodes the agent's behavioral trajectories $\{(o_t^i, a_t^i)\}$ into an embedding space, in which different policies generate different representations. Then, we manually select 15 distinct policies and randomly split them into 8 training policies and 7 testing policies for each scenario. We illustrate the policy representations in Fig. 9. During training or testing, in each episode, we sample one policy from the policy set and equip a team of agents with the policy. Then, we replace one random agent in the team with the ad hoc agent to construct an ad hoc team.

**Save the City.** This is a $14 \times 14$ grid world resource allocation task presented in (Iqbal et al., 2020). In this task, there are 3 distinct types of agents, and their goal is to complete the construction of all buildings on the map while preventing them from burning down. Each agent has 8 actions: stay in place, move to the next grid in one of the four cardinal directions, put out the fire, and build. We set the agents to get a team reward of 100 if they have completed a building and be penalized for -500 when one building is burned down. Agent types include firefighters (20x speedup over the base rate in reducing fires), builders (20x speedup in the building), or generalists (5x speedup in both as well 2x speedup in moving ). Buildings also consist of two varieties:

fast-burning and slow-burning, where the fast-burning buildings burn four times faster. In our experiments, each agent can only access the environmental information within four grids nearby. The observation of each agent contains the information (type, position, and the completion procedure of the building) of each entity in the environment. If one entity is not in the sight range of the agent, the observation vector of this entity would be filled with $0$. We adopt three different scenarios here to verify all methods. They are 2 agents and 1 buildings (2a2b), 4 agents and 3 buildings (4a3b), 6 agents and 4 buildings (6a4b). Similar to Predator Prey, we also manually selected 15 teammates' policies for each scenario. We illustrate their policy representations in Fig. 9. Furthermore, since each team has a random combination of agent types, teammates' behaviors on this environments show a greater diversity. In addition, all buildings' types are randomly assigned in each episode.

## B   LIMITATIONS AND FUTURE WORKS

In this paper, the experiments were carried with relatively small teams (2-8 agents) and with agent behavior largely coming from pre-trained RL policies. In this case, the diversity of teammates' policies can be easily verified by visualizing the policy representations in latent space. However, with the team size increasing, there might be more complex and diverse teamwork situations. This leads to a higher requirement for the training speed and the scalability of the architecture. In addition, the diversity of the policy sets for training has considerable influence on the performance. However, the characterization and quantization of such an influence remain unexplored. We leave these issues for future work.

