# OpenReview forum: "Online Ad Hoc Teamwork under Partial Observability"
_ICLR.cc/2022/Conference — ICLR 2022 Poster_

### Official Review · Reviewer_X2GL · 2021-11-03

**Correctness:** 4
**Technical Novelty And Significance:** 3
**Empirical Novelty And Significance:** 3
**Recommendation:** 8
**Confidence:** 4

**Details Of Ethics Concerns:**

The paper does not discuss limitations or societal impact. The problem of ad hoc teamwork, however, may have societal implications since one would expect that the ability to engage in ad hoc teamwork will be fundamental in agents that cooperatively interact with humans in joint tasks. The proposed approach (requiring full observability during training) may raise privacy issues in settings of human-agent teamwork. Although I believe that we are not yet at a point of having human-agent teamwork scenarios where the proposed approach may concerns, it is nevertheless important to not completely ignore these issues.

**Main Review:**

Overall, I liked the paper. The proposed approach strikes me as novel in the context of ad hoc teamwork, although the idea of building multiple encoder-decoder models that learn a common latent representation from diverse sources of information by joint training through the addition of mutual-information loss terms has been explored in the context of multimodal representation learning (see, for example, references [a, b] below). It is also clearly different from the "main trend" on ad hoc teamwork, which mostly disregards partial observability.

At the same time, there is one aspect about the paper that does not fully convince me and which could be further discussed. My main objection is the assumption that---during training---the agent has full access to the other agent's and the environment's state. I'm fully aware that previous approaches (such as PLASTIC or ConvCPD) assume a "training phase" whereby the ad hoc agent is allowed to learn about the teammates. However, the interaction between the ad hoc agent and teammates is the same during training and testing---something which, in my view, is key to the ad hoc teamwork formulation: the ad hoc agent builds on its interaction with previous teams to quickly adjust to the current (unknown) team. The notion that the ad hoc agent, during training, is allowed full access to the teammates seems more in line with other settings---such as MARL or, perhaps, transfer in MARL---than with ad hoc teamwork.

Refs.

[a] M. Suzuki, K. Nakayama, Y. Matsuo. "Joint multimodal learning with deep generative models." arXiv:1611.01891, 2016.
[b] H. Yin, F. Melo, A. Billard, A. Paiva. "Associate latent encodings in learning from demonstrations." AAAI, 2017.

**Summary Of The Paper:**

The paper addresses the problem of ad hoc teamwork, where an agent learns how to coordinate with a team of unknown teammates. Previous works typically assume that the teammates belong to one of a finite set of possible (known) types; at test time, the ad hoc agent uses the observed teammate behavior to identify their corresponding type and act accordingly.

The proposed method, instead, builds a predictor of the ad hoc agent's marginal utility (which depends on the teammates) that relies on local information alone. The predictor takes as input the ad hoc agent's current observations, $b_t$, and a latent vector, $c$, representing the current "teamwork situation". The teamwork situation representation, $c$, roughly plays the role of "teammate types" in previous works. To train the predictor, the proposed method makes use of an encoder-decoder model that, at training time, learns the "teamwork situation" representation $c$ using full state-action information. The output of the model is used in an "integration network" that estimates the $Q$-function for the underlying MMDP from $c$ and the agent's estimated marginal utility, $u_t$. Finally, since at test time the agent cannot access the full state-action information (and hence cannot estimate $c$ using the aforementioned encoder-decoder model), during training it learns an proxy encoder that builds an estimate $z$ for $c$ using only local information. The overall architecture thus includes two encoder-decoder models jointly trained so that they estimate a coherent representation for the "teamwork situation"---the first using only local information, and the second using full state-action information. At test time, the agent can use the estimated representation, $z$, as input to its marginal utility predictor and act accordingly.

The paper tests the proposed approach in 3 domains (modified coins environment, predator-prey, save the city) showing positive results over existing architectures and a well-established MARL approach (QMIX), and also presents a brief ablation study establishing the relevance of using the two encoder-decoder models to the overall performance of the method.

**Summary Of The Review:**

I believe the paper is interesting and provides a novel contribution in terms of ad hoc teamwork (addressing partial observability). It would be good if additional discussion could be made regarding the assumption of "centralized training".

---

> ### Author Response · Authors · 2021-11-19
> **Response to Reviewer X2GL**
>
> Thank you for your appreciation and useful advices to our work.
>
> **Q1. Why Centralized training and Decentralized Execution (CTDE)?**
>
> A1. CTDE has not been commonly used in training an ad hoc agent yet, because in a conventional setting we do not have the partial observability issue. In other words, the ad hoc agent has full observability during both training and testing, thus the interaction can be same during training and testing. However, with partial observability, the ad hoc agent only receives limited information during execution. This encourages us to design ODITS, aiming to bridge the gap between training and testing. Moreover, the architecture of ODITS can be viewed as a special utilization of CTDE, in the sense that ODITS trains only the ad hoc agent, while in MARL all teammates are trained together.

---

### Official Review · Reviewer_v5Yz · 2021-11-04

**Correctness:** 3
**Technical Novelty And Significance:** 3
**Empirical Novelty And Significance:** 3
**Recommendation:** 6
**Confidence:** 4

**Main Review:**

Overall this is a pretty cool paper. The core insight, which is learning a flexible and adaptable way of utilizing teammate observation in ad hoc teamwork via latent representations seems to be a very useful approach, and the overall architecture seems sound. I do have some comments and questions, however:

- The relationship between the proxy encoder-decoder and the fully-observable encoder-decoder is not entirely clear to me, and it's not entirely obvious to me how robust the current formulation is. There's also the concern that the entire framework might be very difficult to train for even moderately sized environments such as half field offense.

- To be fair, in practice, most ad hoc teamwork approaches, even if they formally assume a finite set of teammate types, handle variable or novel teammates by learning a distribution over previously modeled types (this can be framed as a k experts problem, or a multi-armed bandit, among other approaches). While AATEAM arguably does this implicitly via the attention mechanism it would still be of interest to consider a simpler baseline whose situational encoder is a distribution over existing learned representations per type - how much does the encoder-decoder approach actually buys you?

- It is unclear to me that the approaches utilized in Section 5.2 to engender variability among teammate types actually gets you this - what concrete measures are there to show that there's real variability in single agent policies learned by varying seeds in QMIX, for instance? One would argue that in the very least a separate effort should be put into maximizing variability, or at least promoting it actively.

- I concede that I did not fully understand the take home message in Section 5.4, is it meant to show that simply following single agent QMIX policy works poorly for an ad hoc agent? That does not seem like a very strong result to show. I would argue that a better experiment would be using  a single agent learning algorithm (say PPO or A3C) and showing that it does worse than the ad-hoc policy learning approach (I would very much expect it to, but at least I understand what that experiment aims to show).


**Summary Of The Paper:**

In this paper the authors tackle the challenge of ad hoc teamwork with unknown teammate types. Unlike most previous work, the proposed framework, ODITS, does not assume some finite set of teammate types or full observability, and adapts to current teammates via the utilization of a teammate situation encoder-decoder framework, which learns a latent representation of the teammate configuration the agent is currently observing (this observation can be partial). The authors subsequently show this approach outperforms state of the art baselines such as AATEAM on a several domains, namely a modified version of coin game, predator prey, and save the city.

**Summary Of The Review:**

Overall this is a reasonably solid paper which presents a novel approach for ad-hoc teamwork policy learning via learning latent embeddings of partial teammate observations.

---

> ### Author Response · Authors · 2021-11-19
> **Response to Reviewer v5Yz**
>
> We thank the reviewer for your constructive feedback.
>
> **Q1. The relationship between the proxy encoder-decoder and the fully-observable encoder-decoder.**
>
> A1. ODITS follows the common Centralized Training and Decentralized Execution (CTDE) [1] framework, where the teamwork situation encoder-decoder accesses the full environment information during training while the proxy encoder-decoder works during execution under partial observability. So we expect the proxy encoder-decoder to capture information which is consistent with the teamwork situation encoder-decoder by introducing the mutual information learning objective. We examine the robustness of them by a series of ablations in section 5.5： **w/o info.** indicates that the mutual-information loss function is the key to promote the proxy encoder-decoder to generate effective adaptive behaviors; **w/o infer.** indicates that the proxy encoder-decoder can reconstruct the core information of teamwork situation from the partial observations; **w/o integ.** indicates that the fully-observable encoder-decoder indeed can capture the information of teamwork situation which determines the current adaptive behaviors of the ad hoc agent. Indeed we only test ODITS in environments with a relatively small number of agents (e.g., 8 predators and 4 preys), since the diversity of teammates’ policies can be easily verified in such cases. We leave the setting of large number of teammates to future work, as discussed in Appendix B.
>
> **Q2. It would still be of interest to consider a simpler baseline whose situational encoder is a distribution over existing learned representations per type.**
>
> A2. We agree that there might be some simpler baselines. Actually, we also compared ODITS with other two works: **PLASTIC** and **ConvCPD**. Specifically, they both have a belief model to predict the belief distribution over all teammates' types. The experimental results (Fig.4) show that these simpler baselines always get worse performance than our model, indicating that the encoder-decoder architecture indeed improves the ad hoc teamwork performance.
>
> **Q3. What concrete measures are there to show that there's real variability in single agent policies learned by varying seeds in QMIX, for instance?**
>
> A3. We find it is hard to directly measure the divergence of agent policies. Instead, we use various tricks to obtain different policies and visualize the selected policies in Figure 9, Appendix A.2 to ensure diversity. Specifically, we utilized 5 algorithms (VDN, MAVEN, ROMA, QMIX, IQL) to train the candidate policies of teammates using their open-source implementations based on PyMARL. We used 4 different random seeds for each algorithm and saved the corresponding models at 3 different training steps (3 million, 4 million, and 5 million). As a result, we get 60 candidate teammates’ policies for each scenario. To ensure diversity of cooperative behaviors, we visualize policy representations of all candidate models by using the open-source implementation mentioned in [2]. This mechanism encodes the agent’s behavioral trajectories into an embedding space, in which each policy representation can reconstruct the corresponding policy. Then, we manually select 15 distinct policies and randomly split them into 8 training policies and 7 testing policies for each scenario.
>
> **Q4. I would argue that a better experiment would be using a single agent learning algorithm (say PPO or A3C) and showing that it does worse than the ad-hoc policy learning approach.**
>
> A4. Section 5.4 aims to show that traditional MARL methods are not suitable for solving ad hoc problems. Regarding the single agent learning, the baseline "Combined" in Section 5, Fig.4 is actually a DQN implementation of single agent learning. During training, we consider other teammates a part of the environment dynamics. The experimental results show that the DQN algorithm has poor performance in ad hoc teamwork. Other learning algorithms (e.g., PPO, A3C) may have different performances. However, intuitively, without mechanisms to handle non-stationary or previously unseen teammates, single agent learning often fails to learn effective policies in the ad hoc setting.  This has also been demonstrated in literatures [3,4].
>
> [1] F. A. Oliehoek, M. T. Spaan, N. Vlassis, and S. Whiteson. Exploiting locality of interaction in factored dec-pomdps. In AAMAS, 2008.
>
> [2] R Raileanu, Goldstein, M. , Szlam, A. , & R Fergus.  Fast adaptation via policy-dynamics value functions. In ICML, pages 7078--7089, 2020.
>
> [3] Rahman, Muhammad A., et al. "Towards Open Ad Hoc Teamwork Using Graph-based Policy Learning." In ICML, 2021.
>
> [4] Barrett, Samuel, and Peter Stone. "Cooperating with unknown teammates in complex domains: A robot soccer case study of ad hoc teamwork." In AAAI. 2015.

---

### Official Review · Reviewer_7i5n · 2021-11-06

**Correctness:** 4
**Technical Novelty And Significance:** 2
**Empirical Novelty And Significance:** 3
**Recommendation:** 6
**Confidence:** 4

**Main Review:**

**Strengths**

Ad-hoc teaming is an important problem from the perspecitve of real world applications. Removing some of limitations of required prior knowledge is quite important to allow full data-driven learning of such behavior. Paper's clear focus on this problem leads to integration of many ideas from latent-variable generative modeling in the generative modeling community with multi-agent RL solution techniques in the ODITS framework. The general idea of distilling high-level semantics of teammate behavior into a latent vectors makes sense and has been explored in some of the related works, but doing that in an online manner remained unexplored. Other ideas about using a hypernetwork for integration (as done by QMIX) and utilizing the CTDE assumption to its fullest while learning a proxy encoder for decentralized execution are all well utilized. There is a clear train-test agent separation protocol in the expriments and comparisons with multiple baselines and different numbers of agents and types.

**Weaknesses**

Fig 3 fill color seems to suggest something is wrong with AATEAM. In RL the failures are seldom so consistent that unlearning would have such a low spread around the mean. Techniques using some form of bi-level optimization or diversity should be relevant related works [1,2].
It's understandable that environments with more complicated dynamics were used given the number of training iterations required for the experiments (which would have been slower too because of additional computation added), it would be useful to add a limitations section to the paper. The experiments were still with relatively small teams and with agent behavior largely coming from pre-trained RL policies. How different the RL policies were might be hard to really understand but it might be useful to report action distribution divergences on pre-selected trajectories along with the t-SNE plot. Overall it's still difficult to say if the selected policies for testing are representative enough as say doing human evaluation where the style of play might be completely different.

[1] https://openreview.net/forum?id=Bkl5kxrKDr
[2] https://arxiv.org/abs/1907.03840




**Summary Of The Paper:**

The paper focuses on the important problem of building agents that can function as useful teammembers in a ad-hoc team where agents can change behavior over an episode. The paper propose a new framework named ODITS that allows training an agent for ad-hoc teaming applications without strong assumptions on observability or pre-defined roles and categories. It does so my learning a latent variable representation of teammate behaviors in a data-driven fashion during policy training. The framework also adds an information-based regularizer to allow learning these variables from partial local observations of an agent (using all available information during training in a CTDE manner). The paper performs experiments on three different environments to demonstrate their outperformance.


**Summary Of The Review:**

Overall it's a good combination of exisiting ideas in latent-variable modeling and CTDE MARL to demonstrate effective ad-hoc teaming behavior.

---

> ### Author Response · Authors · 2021-11-19
> **Response to Reviewer 7i5n**
>
> Thank you for your useful suggestions!
>
> **Q1. Fig 3 fill color seems to suggest something is wrong with AATEAM.. unlearning would have such a low spread around the mean.**
>
> A1. We have carefully checked the details of the experiment and found two possible reasons: 1）The stochasticity of the modified coin game environment is low, in the sense that the policies of teammates and the initial positions of coins are all deterministic. 2) The AATEAM is more like a supervised learning framework instead of a reinforcement learning framework, so the random seeds might have small influence on the performance of AATEAM. We hope that it clears your concerns.
>
> **Q2. Techniques using some form of bi-level optimization or diversity should be relevant related works.**
>
> A2. Generating diverse policy set is indeed beneficial for training ad hoc agents. We will cite and discuss about these  works in the new version of this paper.
>
> **Q3. It would be useful to add a limitations section to the paper.**
>
> A3. We have added a paragraph in Appendix Section B to discuss the limitations and future works, including the size of the team and the diversity of teammates’ policies.
>
> **Q4. It might be useful to report action distribution divergences on pre-selected trajectories along with the t-SNE plot.**
>
> A4. We have considered to report action distribution divergence along some trajectories but found that it depends heavily on the initial states being selected. Therefore, we might lose global perspective on how the policies diverge. Instead, we think that the t-sne plot of policy representations is a more concise way to show and visualize the divergence of policies, as is proposed in [1] . The policy representation encoder takes the trajectories collected by the given policy as input, and outputs the policy representation which can predict the policy's decision if the state is given. As a result, if any two policy representations are distant from each other in the representation space, they tend to choose different actions under the same state.
>
> [1] R Raileanu, Goldstein, M. , Szlam, A. , & R Fergus.  Fast adaptation via policy-dynamics value functions. In ICML, pages 7078--7089, 2020.

---

### Official Review · Reviewer_Z8fc · 2021-11-06

**Correctness:** 3
**Technical Novelty And Significance:** 3
**Empirical Novelty And Significance:** 3
**Recommendation:** 6
**Confidence:** 2

**Main Review:**

Just as a forewarning, I'm not highly familiar with this problem so what I say should be taken with a grain of salt, but here are my observations nonetheless.

### Originality

It seems like the main claims to originality verge on the fact that ODITS:
1) doesn't require pre-defining agent types
2) is able to adapt in an online fashion to changing and unseen agents/situations
3) is able to deal with partially observable environments

Two works which are not cited, seem to potentially be related (and check at least some of the boxes above) are:
- Deep Interactive Bayesian Reinforcement Learning via Meta-Learning. From the paper: "A popular way of modelling other agents is type-based modelling which assumes that the other agent has one of several types that can be pre-defined by an expert, or learned from data [2, 5, 47]. In this paper, we learn a latent variable model, in which the latent variables can be seen as a continuous representation of agent types learned in an unsupervised way via interaction.". My understanding is that this would check boxes 1 and 2, and might be extendible to also achieve 3.
- Georgios Papoudakis and Stefano V Albrecht. Variational autoencoders for opponent modeling in multi-agent systems. My understanding is that in this paper they focus on partial observability (3) potentially at the expense of adaptation (2).

In any case, it would be helpful to add these to the related work and address what the main differences are. If particularly related, they could also constitute additional baselines for the method. MeLIBA in particular seems like a potentially natural choice of baseline to the method presented.

### Clarity

The paper is well written and overall quite clear. Some minor suggestions/questions:
- Figure 1: it's unclear what the groups on the right hand side are supposed to represent. For example, why are there 3 green agents and 5 yellow ones? Is this just supposed to represent the proportions with which such agents are sampled?
- In the "Marginal Utility" heading, it might be useful to briefly give a definition of Marginal Utility, rather than just referring to other work. I'm not sure how common this term is in this subfield – it might be standard enough for this to be reasonable.
- In various places throughout the text, you use the expression "key insight". It might be useful to vary this expression or use it more sparingly for the thing that is actually most "key" – especially the usage in the introduction seems quite underwhelming.
- Figure 3: I would separate the legend (containing "Teammate", "Other agent", and "Coins") from the training and testing types. I found the illustration quite confusing.
- Section 5.1. Partial observability means that the main agent might not see what coins the teammate has taken. I would describe qualitatively what the main agent does in order to address this? Do they try to find the teammate as soon as possible and then follow them? Do they scan the entire grid until they are certain of what coins the teammate took? Neither?

I appreciated the use of ablations in Sec. 5.5 showing the relative importance of the various components of the architecture.

**Summary Of The Paper:**

This paper proposes a new method, ODITS, to enable effective ad hoc cooperation without requiring to predefine a static set of teammate types. This is done by training encoder networks to extract latent variables which capture the "situation type", which is trained to be a sufficient statistic for predicting the joint action value. This enables the agents to adapt in online fashion (it doesn't assume static other agents), and additionally the methodology is able to deal with partially observable environments.

**Summary Of The Review:**

I don't feel qualified to comment on the method's limitations in the broader context of previous work. Outside of the method's quality, the experiments seem well executed and the work is relatively clear. It seems like the method presented does better than the baselines considered, although these might be limited (as mentioned in "originality"). Overall, I'll mostly remit to other reviewer's judgement on the method, and wouldn't be opposed to this paper being accepted.

---

> ### Author Response · Authors · 2021-11-19
> **Response to Reviewer Z8fc**
>
> Thank you for your kind suggestions and helpful feedback!
>
> **Q1. Comparison with “Deep Interactive Bayesian Reinforcement Learning via Meta-Learning”.**
>
> A1. The idea of the referred paper is to condition the ad hoc agent’s policy on a belief over teammates, which is updated following the Bayesian rule. Yes it checks box 1 because the belief is defined over the actions of teammates, instead of teammates’ types. It also allows online adaptation to non-stationary teammates, which matches the setting of ad hoc problems. However, it does not check the second part of box 2, i.e., adapt to unseen agents/situations, because you only have belief of the agents you have interacted with. Learning belief of new agent from scratch costs more interactions, which might be expensive. Furthermore, extending MeLIBA to the partial observability setting means that the agent has to infer teammates’ actions from only its local observations. This can be very challenging due to limited information of local observations. Our ablation study also indicates the importance of the aid from global information: removing the mutual information loss significantly influence the performance of ODITS, as shown in Figure 6, w/o info.
>
> Comparing with our work, the key difference is that we learn representations of teamwork situations, instead of that of each individual agent. This allows us to relax many unnecessary assumptions, including agent types/numbers, prior beliefs and full observabilities. Therefore, our framework is more general and adapts to various situations. However, we do believe this work is relevant and will discuss about it in the new version.
>
> **Q2. Comparison with “Variational Autoencoders for Opponent Modeling in Multi-Agent Systems”.**
>
> A2. This paper proposes to use VAE to learn fixed-policy opponents. The proposed VAE is similar to the proxy encoder in ODITS. However, in the ad hoc setting, the opponents/teammates might be non-stationary and even previously unseen, where a simple VAE would fail to learn. In ODITS, we address this issue by introducing the teamwork situation encoder/decoder and train them jointly with the proxy encoder/decoder. Our ablation study also emphasizes the importance of the teamwork situation encoder/decoder, as is shown in Figure 6, w/o integ. We will also add discussions on this work in the new revision.
>
> **Q3. Figure 1: it's unclear what the groups on the right hand side are supposed to represent.**
>
> A3. The reason why agents with different colors in Fig.1 have different group sizes is that we expect to show that in the ad hoc teamwork problem teammate policies are diverse, so they might be suitable for various group sizes. We have added illustrations in the new version.
>
> **Q4. Other clarity issues including: definition of marginal utility, reducing the expression of the key insight, legend of Figure 3, description of the modified coin game.**
>
> A4. Thank you for the valuable suggestions. We have addressed all of these issues in the new version.

---

### Author Response · Authors · 2021-11-19
**SUMMARY OF UPDATES**


We thank all the reviewers for their time spending on our paper and insightful feedbacks. We have updated the paper according to the suggestions of the reviewers. Here is the summary of the updates:

1.	We have added some descriptions in Section 2 to discuss two related works in the field of agent modeling [1, 2].
2.	We have added some descriptions about Fig.1 in Section 3 to prevent confusion -- the reason why agents with different colors in Fig.1 have different group sizes is that we expect to show that in the ad hoc teamwork problem teammate policies are diverse, so they might be suitable for various group sizes.
3.	We have added illustration on the definition of Marginal Utility in Section 3.
4.	We have reduced the frequent expression of  "key insight" throughout the whole paper.
5.	We have altered the illustration of modified coin game in Fig.3 and Fig.8 to prevent confusion – We have separated the legend (containing "Teammate", "Other agent", and "Coins") from the training and testing types.
6.	We have added some descriptions in Section 5.1 to reveal that the aim of the ad hoc agent in the modified coin game is to infer its current teammates’ desired coins and collect them as much as possible.
7.	We have cited and discussed techniques using some form of bi-level optimization or diversity in Section 2 [3, 4].
8.	We have added a paragraph in Appendix Section B to discuss the limitations and future works, including the size of the team and the diversity of teammates’ policies.

[1] L. Zintgraf, S. Devlin, K. Ciosek, S. Whiteson, and K. Hofmann. Deep interactive bayesian reinforcement learning via meta-learning. arXiv preprint arXiv:2101.03864, 2021.

[2] G. Papoudakis and S. V. Albrecht. Variational autoencoders for opponent modeling in multi-agent systems. In AAAI Workshop on Reinforcement Learning in Games, 2020.

[3] P. Muller, S. Omidshafiei, M. Rowland, K. Tuyls, J. Perolat, S. Liu, D. Hennes, L. Marris, M. Lanctot, E. Hughes, Z. Wang, G. Lever, N. Heess, T. Graepel, and R. Munos. A generalized training approach for multiagent learning. In International Conference on Learning Representations, 2020.

[4] R. Canaan, J. Togelius, A. Nealen, and S. Menzel. Diverse agents for ad-hoc cooperation in hanabi. arXiv preprint arXiv:1907.03840, 2019.

---

### Decision · Program_Chairs · 2022-01-20

**Decision:**

Accept (Poster)

**Comment:**

The paper presents a method for cooperative ad-hoc collaboration by learning latent representations of the teammates. The method is evaluated in three domains. All the reviewers agree that the method is novel and adds an interesting contribution to the important and difficult problem of the ad-hoc collaboration, making fewer assumptions about the team and the teammates.

The next version of the paper should comment:

- On the societal impact of the centralized training.
- Wang et al, CoRL 2020, https://arxiv.org/abs/2003.06906, which addresses the cooperative tasks in the ad-hoc teams without privileged knowledge and assumptions about the teammates.